# Cerebellar modules operate at different frequencies

**Haibo Zhou[1][†], Zhanmin Lin[1][†], Kai Voges[1], Chiheng Ju[1], Zhenyu Gao[1], Laurens WJ Bosman[1], Tom JH Ruigrok[1], Freek E Hoebeek[1], Chris I De Zeeuw[1,2]\*[‡], Martijn Schonewille[1]\*[‡]**

[1]Department of Neuroscience, Erasmus MC, Rotterdam, Netherlands; [2]Cerebellar Coordination and Cognition, Netherlands Institute for Neuroscience, Amsterdam, Netherlands

**Abstract** Due to the uniform cyto-architecture of the cerebellar cortex, its overall physiological characteristics have traditionally been considered to be homogeneous. In this study, we show in awake mice at rest that spiking activity of Purkinje cells, the sole output cells of the cerebellar cortex, differs between cerebellar modules and correlates with their expression of the glycolytic enzyme aldolase C or zebrin. Simple spike and complex spike frequencies were significantly higher in Purkinje cells located in zebrin-negative than zebrin-positive modules. The difference in simple spike frequency persisted when the synaptic input to, but not intrinsic activity of, Purkinje cells was manipulated. Blocking TRPC3, the effector channel of a cascade of proteins that have zebrin-like distribution patterns, attenuated the simple spike frequency difference. Our results indicate that zebrin-discriminated cerebellar modules operate at different frequencies, which depend on activation of TRPC3, and that this property is relevant for all cerebellar functions.

**\*For correspondence:**
c.dezeeuw@erasmusmc.nl (CIDZ)
m.schonewille@erasmusmc.nl
(MS)

[†]These authors contributed
equally to this work

[‡]CIDZ and MS share senior
authorship

**Competing interests:** The
authors declare that no
competing interests exist.

**Reviewing editor**: Dora E
Angelaki, Baylor College of
Medicine, United States

## Introduction

Resolving structure–function relations remains one of the main challenges of modern neuroscience. The unique cyto-architecture of the cerebellum is characterized by the crystalline matrix of its sagittally oriented PC dendrites and climbing fibers and its orthogonally running parallel fibers (*Larsell, 1972*; *Voogd, 2011*). The ubiquitous nature of this relatively simple matrix throughout all lobules and modules of the cerebellar cortex made scientists predict in 1967 that this neuronal machine was probably the first to be elucidated (*Eccles, 1967*). Yet, about half a century later, we have collected a wealth of information about the molecular and physiological identity of the various cell types in the cerebellum (*Gao et al., 2012*), but gross structure–function relations are still largely lacking. For example, the amount of evidence for physiological differences within the cerebellar cortex is limited and there is little comparative analysis of spiking activity throughout the cerebellar cortex in adult awake animals. In fact, even most slice physiology studies do not discriminate between lobules or modules, indicative of the fact that the cerebellum is still considered physiologically homogeneous.

At the same time, several molecular markers have been identified that can subdivide the cerebellar cortex into distinct bands (*Apps and Hawkes, 2009*). The best-known of these molecules are the zebrins, which are highly expressed by specific bands of Purkinje cells (PCs), that is the sole output of the cerebellar cortex. Immunostainings for both zebrin I and II give rise to symmetric stripes that are oriented perpendicular to the cerebellar folds (*Brochu et al., 1990*; *Leclerc et al., 1990*). The combined presence of zebrin-positive and zebrin-negative PCs can be found in all vertebrate classes, and zebra-like patterns are present in the cerebellum of birds and mammals, varying from pigeons and mice up to monkeys and humans (*Brochu et al., 1990*; *Sillitoe et al., 2003*; *Marzban and Hawkes, 2011*; *Graham and Wylie, 2012*). In most cases, adjacent PCs with zebrin II (from hereon referred to as zebrin) are located in the same bands, receive CF inputs from the same part of the inferior olive, and

**eLife digest** The cerebellum, located at the back of the brain underneath the cerebral hemispheres, is best known for its role in the control of movement. Despite its small size, the cerebellum contains more than half of the brain's neurons. These are organized in a repeating pattern in which cells called Purkinje cells receive inputs from two types of fibers: climbing fibers, which ascend into the cerebellum from the brainstem; and parallel fibers, which run perpendicular to the climbing fibers. This gives rise to a characteristic 'crystalline' structure.

As a result of this uniform circuitry, it was widely believed was that all Purkinje cells throughout the cerebellum would function the same way. However, the presence of distinct patterns of gene expression in different regions suggests that this is not the case. Molecules called zebrins, for example, are found in some Purkinje cells but not others, and this gives rise to a pattern of zebrin-positive and zebrin-negative stripes. A number of other molecules have similar distributions, suggesting that these differences in molecular machinery could underlie differences in cellular physiology.

Zhou, Lin et al. have now provided one of the first direct demonstrations of such physiological differences by showing that zebrin-positive cells generate action potentials at lower frequencies than zebrin-negative cells. This pattern is seen throughout the cerebellum, and is evident even when the positive and negative cells are neighbors, which indicates that these differences do not simply reflect differences in the locations of the cells or differences in the inputs they receive from parallel fibers. Additional experiments revealed that the distinct firing rates are likely not generated by zebrin itself, but rather by proteins that are expressed alongside zebrin, most notably those that work through an ion channel called TRPC3.

By showing that cells arranged in the same type of circuit can nevertheless have distinct firing rates, the work of Zhou, Lin et al. has revealed an additional level of complexity in the physiology of the cerebellum. In addition to improving our understanding of how the brain controls movement, these findings might also be of interest to researchers studying the increasing number of neurological and psychiatric disorders in which cerebellar dysfunction has been implicated.

project their axons to the same part of the cerebellar nuclei (*Voogd and Ruigrok, 2004*; *Pijpers et al., 2006*; *Sugihara and Shinoda, 2007*; *Apps and Hawkes, 2009*; *Sugihara et al., 2009*; *Sugihara, 2011*). Moreover, although their various terminal rosettes may be located in different parts of the cerebellar cortex, individual mossy fibers often also adhere to the same zebrin signature (*Pijpers et al., 2006*). As such zebrin may be regarded as a biomarker linking different cerebellar cortical zones, potentially binding activity of different olivo-cerebellar modules and mossy fiber systems (*Voogd and Ruigrok, 2004*; *Pijpers et al., 2006*; *Ruigrok, 2011*). However, what the basic characteristics of this zebrin-related activity might be is unknown. Since zebrin has been identified as the glycolytic enzyme aldolase C, its presence might in principle be linked to the level of metabolic and/or electrophysiological activity. Indeed, the distribution of zebrin in the cerebellum is similar to that of the excitatory amino acid transporter 4 (EAAT4) and complementary to splice variant b of the metabotropic glutamate receptor 1 (mGluR1b) (*Dehnes et al., 1998*; *Mateos et al., 2001*; *Wadiche and Jahr, 2005*). Intracellularly, several proteins in a molecular cascade linked to mGluR1 are also expressed in zebrin-like bands, including the IP3-receptor (*Furutama et al., 2010*), PLCβ3/4 (*Sarna et al., 2006*), PKCδ (*Barmack et al., 2000*), and NCS-1 (*Jinno et al., 2003*). This cascade controls the activity of the transient receptor potential cation channel type C3 (TRPC3) (*Hartmann et al., 2008*), which in turn can influence the firing activity of PCs (*Sekerkova et al., 2013*). We hypothesized that differential activity of this cascade of proteins with zebrin-related expression might lead to differential activity of their effector channel, TRPC3, and thereby to differences in simple spike (SS) firing frequency between modules (*Kim et al., 2012a*, *2012b*).

To test this hypothesis, we investigated the activity of PCs in awake mice at rest in relation to the zebrin-identity of their module. We demonstrate that there are zebrin-related differences in firing frequency of both SSs and complex spikes (CSs), that these differences are intrinsically driven, and that they are consistently present throughout the cerebellar cortex contributing to all its functions.

## Results

### Simple spike firing activity differs between Purkinje cell populations

We performed extracellular recordings from PCs in the cerebellar cortex of awake, restrained C57Bl/6 mice at rest with the use of double-barrel electrodes marking the recording location with Alcian Blue (*Figure 1A,B*). Purkinje cells were identified by the presence of SSs and CSs and the consistent presence of a pause in SS firing after a CS (i.e., climbing fiber pause) (*Figure 1C–F*). Recordings that were used for analysis had to meet several criteria including a minimum duration of 120 s, stable spike amplitude over the whole recording period and no detectable tissue damage in a 400-µm radius

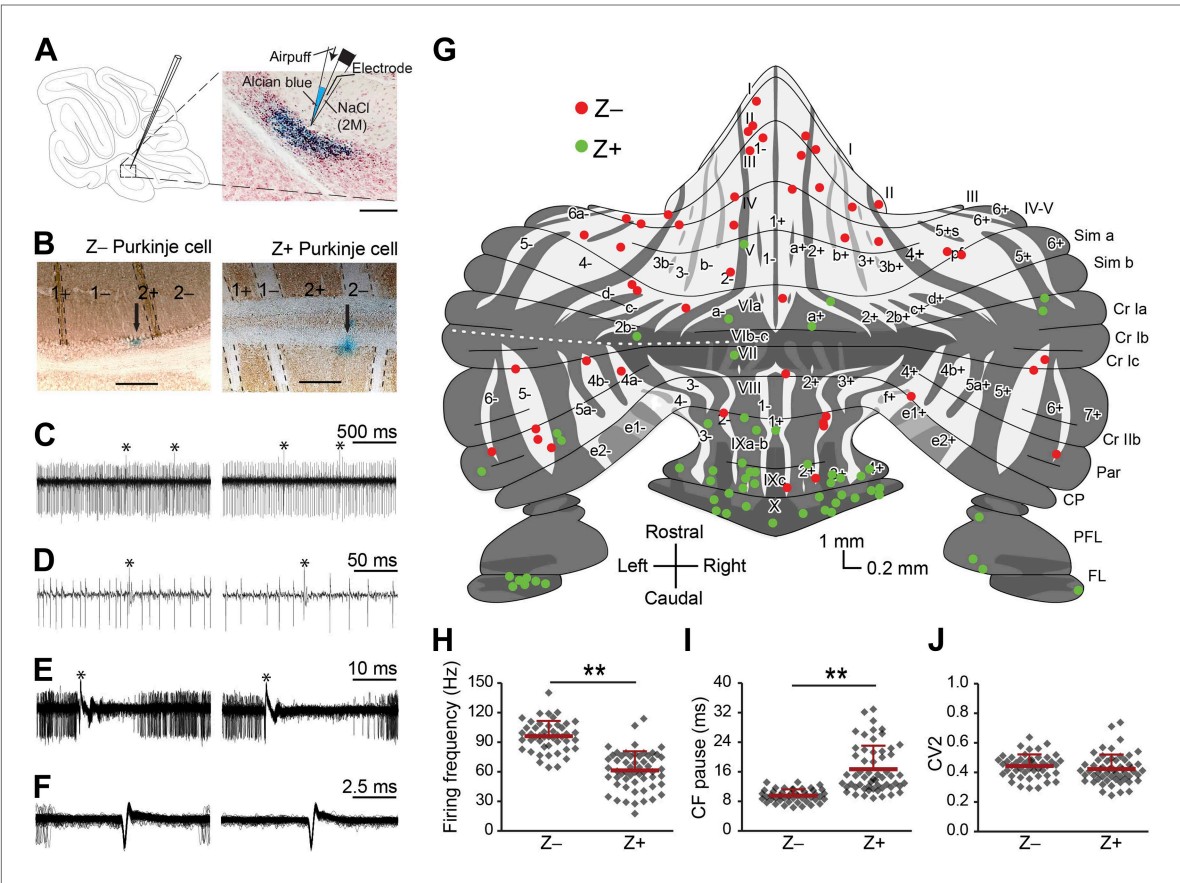

**Figure 1**. Simple spike firing activity differs between Purkinje cell populations. (**A**) Extracellular recordings were made from PCs in the cerebellar cortex of awake mice, using double barrel glass electrodes (right). Dye injections were placed to histologically identify the recording location. (**B**) Photomicrographs of coronal sections with examples of zebrin-negative (Z−, left) and zebrin-positive (Z+, right) identified Purkinje cells in lobule II and lobule IX, respectively. Cells are marked by dye injections (blue, indicated by arrows), zebrin is stained brown, dotted lines demark zebrin borders. Note that Z+ stripes in lobules I–III are very narrow. (**C** and **D**) Example trace of Z− and Z+ Purkinje cell recordings identified by its hallmark feature, the occurrence of complex spikes (asterisk) and simple spikes. (**E**) Recordings were confirmed to be from a single neuron by the consistent pause in simple spike firing following each complex spike, in the overlay. (**F**) Overlay of simple spikes. (**G**) Distribution of recorded Z− and Z+ cells throughout the unfolded cerebellar cortex based on zebrin II compartments. (**H**) Simple spike firing frequency is significantly lower in identified Z+ PCs compared to Z− PCs (Z−: n = 47 cells, 26 mice; Z+: n = 57 cells, 34 mice; t = 9.942, p<0.001). (**I**) In line with the lower simple spike firing frequency, the climbing fiber pause was longer in identified Z+ Purkinje cells (CF pause; t = −7.482, p<0.001). (**J**) Simple spike regularity is not different between Z+ and Z− PCs (CV2: t = 1.147, p=0.234). Error bars represent SD, *p<0.05, **p<0.001. Schematic drawing in **A** was adapted from *Sugihara and Quy (2007)* with permission. Scale bars in **A** and **B** indicate 100 and 200 µm, respectively.

The following figure supplements are available for figure 1:

**Figure supplement 1**. Experimental approach and histological verification.

**Figure supplement 2**. Stability of key parameters over the recording time.

(see also *Figure 1—figure supplement 1*). Following perfusion of the animals and processing of their cerebella, the zebrin-negative (Z−) and zebrin-positive (Z+) zones were identified by immunostaining. Of the 104 PCs included in the analysis (50 mice), 47 and 57 cells were located in Z− and Z+ zones, respectively (*Figure 1G*, plot adapted from *Sugihara and Quy, 2007*). The SS firing frequency was significantly higher in Z− zones than in Z+ zones (Z−: 96.1 ± 15.4 Hz, Z+: 61.4 ± 19.3 Hz, $t = 9.942$, p<0.001) (*Figure 1H*). In line with this the climbing fiber pause was also significantly longer ($t = −7.482$, p<0.001) in Z+ zones (*Figure 1I*) (see also *Paukert et al., 2010*). Both the SS firing frequency, SS regularity and CS firing frequency were stable over time (*Figure 1—figure supplement 2*). In contrast to the SS firing frequency and climbing fiber pause, the waveform and regularity of SSs did not consistently depend on zebrin identity in that average half-width and mean coefficient of variation for adjacent intervals (CV2) were not significantly different between Z− and Z+ PCs (half-width: $t = −1.133$, p=0.260, data not shown; CV2: $t = 1.197$, p=0.234) (*Figure 1F–J*).

## Simple spike firing frequency correlates with the zebrin identity of Purkinje cells

Due to the heterogeneous distribution of Z+ vs Z− Purkinje cells over the cerebellar cortex, the majority of the Z+ cells were recorded in the posterior half, whereas the Z− cells were predominantly from the anterior half. Hence, one could argue that the difference between Z+ and Z− is related to location, rather than directly linked with zebrin identity. Re-plotting the results, but now color-coded for simple spike frequency to facilitate individual comparisons, seems to largely contradict this possibility (*Figure 2—figure supplement 1*). To more thoroughly test our hypothesis that differences are indeed related to zebrin identity, we also attempted to record neighboring, online identified, Z+ and Z− PCs in a single experiment. To this end, we performed two-photon imaging in vivo in awake, head-fixed mice that express enhanced green fluorescent protein (eGFP) under the EAAT4 promoter in a pattern similar to that of zebrin (*Gincel et al., 2007*). In the dorsal layer of lobule V, VI, and Crus I we identified Z+ and Z− bands and recorded PCs in adjacent zebrin bands (*Figure 2A,B*). In line with our hypothesis, we observed higher simple spike activity in Z− than in Z+ Purkinje cells (Z+: 36.0 ± 15.5 Hz, n = 8; Z−: 75.8 ± 19.5 Hz, n = 9; $t = 4.618$, p<0.001) and concommitant longer climbing fiber pauses (*Figure 2C*). In contrast to the immunohistochemically subdivided PC dataset (*Figure 1*), that covers the entire cerebellar cortex, this spatially restricted dataset did show a difference in simple spike regularity, suggesting that variations in regularity may occur more locally.

Finally, to extend this analysis over the entire cortex, we compared Z+ vs Z− PC activity per transverse zone. Along the rostro-caudal axis the cerebellum can be subdivided into four transverse zones: the anterior, central, posterior, and nodular zone (*Ozol et al., 1999*). We consistently observed a similar difference in simple spike activity between Z+ and Z− PCs in each zone, independent of the location within the cerebellar cortex (*Figure 2D–E*). This approach also revealed a difference within the population of Z+ PCs. Whereas the simple spike firing frequency of Z− PCs is comparable over different transverse zones, Z+ PCs firing rate is lower in the anterior zone when compared to the nodular zone (p=0.018, One-way ANOVA followed by Tukey's post-hoc test).

If the SS activity of PCs depends on the presence of zebrin, one should also observe differences between lobules, as there is a gradual increase in zebrin-positive modules and, thus, average zebrin intensity from lobule I to lobule X in the vermis and the corresponding lobules in the hemispheres (*Sugihara and Shinoda, 2004*) (*Figure 1G*, *Figure 3A*). Indeed, when we extend the immunohistochemically analyzed dataset with recordings from all lobules in which the zebrin identity was not determined to generate one large, randomly sampled dataset (combined n = 245), our prediction is confirmed. Both the firing frequency and climbing fiber pause of SS activity, among the different lobules in the vermis and the hemispheres, show robust and consistent correlations with the averaged intensity of zebrin staining (for firing frequency, vermis: $r = 0.893$, p=0.007; hemisphere: $r = 1.000$, p<0.001; for climbing fiber pause, vermis: $r = −0.929$, p=0.003; hemisphere: $r = −1.000$, p<0.001) (*Figure 3A–C*, *Figure 3—figure supplement 1*). In contrast, the CV2 of SSs could not be consistently related to the zebrin intensity in the vermis and hemispheres (vermis: $r = 0.929$, p=0.003; hemisphere: $r = −0.300$, p=0.624) (*Figure 3D*).

## Complex spike characteristics depend on the zebrin identity

Reduced tonic SS activity of PCs at rest, as observed in Z+ modules, will lead to enhanced activity of the GABAergic neurons in the cerebellar nuclei that inhibit inferior olivary neurons (*Chen et al., 2010*;

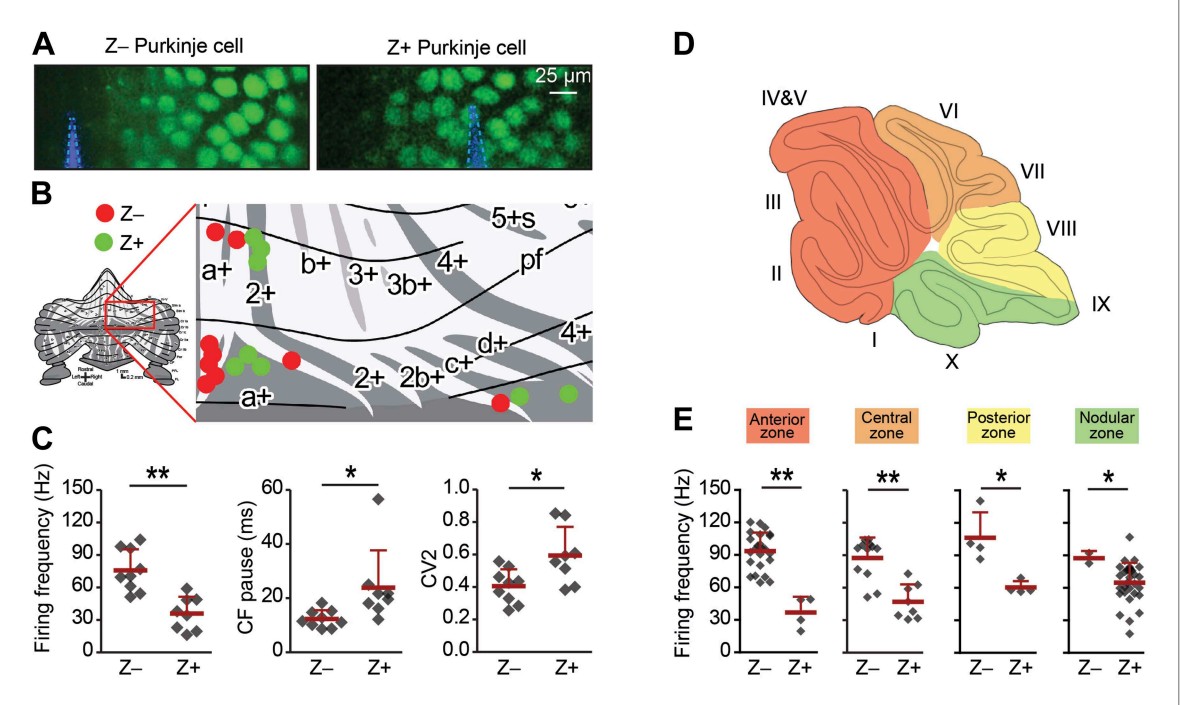

**Figure 2**. Simple spike firing frequency correlates with the zebrin identity of Purkinje cells. To determine of the differences in simple spike activity are related to the location of the Purkinje cells, or to their zebrin identity, we compared PC activity of Z+ against Z− Purkinje cells in various smaller areas of the cerebellar cortex. (**A**) To more directly test the link with zebrin, we used EAAT4-eGFP mice that express eGFP in a pattern similar to that of zebrin. Two-photon images show an EAAT4+/Z+ band (green) in lobule V of an EAAT4-eGFP mouse: left, electrode (blue) positioned in the adjacent negative band; right, electrode (blue) in the positive band. (**B**) The activity of 17 zebrin-identity determined PCs (Z+, n = 8; Z−, n = 9, 5 mice) from lobule V, VI, and Crus I was recorded. (**C**) The difference in simple spike firing frequency was pertained in this subset of Purkinje cell recordings (Z+: 36.0 ± 15.5 Hz; Z−: 75.8 ± 19.5 Hz; *t* = 4.618, p<0.001), indicating that this difference is linked to zebrin identity, rather than lobular location. In contrast to data obtained with immunostaining for zebrin, the regularity of simple spikes also differs in this subpopulation (*t* = −2.715, p<0.016). (**D**) Cerebellar Purkinje cells can be subdivided based on the input they receive into four transverse zones: the anterior (red), central (orange), posterior (yellow), and nodular (green) zone. (**E**) The difference in simple spike firing frequency between Z+ and Z− Purkinje cells is consistently present throughout all transverse zones. In each of the four transverse zones, the simple spike rate was significantly lower in Z+ compared to Z− Purkinje cells (all p<0.05, One-tailed Student's *t* test). Note that simple spike frequency within different Z+ subgroups was also variable, in that the frequency in the anterior zone was lower than that in the nodular zone (p=0.018, One-way ANOVA, followed by Bonferroni's posthoc test). Error bars represent SD, *p<0.05, **p<0.001.

The following figure supplements are available for figure 2:

**Figure supplement 1**. Overview with color-coded simple spike frequency for all identified Z+ and Z− Purkinje cells.

*De Zeeuw et al., 2011*). Therefore, one can expect the CS activity that results from activity in the climbing fibers originating in the inferior olive to be reduced as well. This prediction indeed holds (*Figure 4A*). The CS activity of immunostaining identified Z+ PCs was significantly lower than that in Z− PCs (same PCs as described in *Figure 1G–J*; Z−: 1.13 ± 0.25 Hz, Z+: 0.92 ± 0.28 Hz, *t* = 3.926, p<0.001). This difference persisted in the subset of two-photon imaging identified Z+ and Z− PCs recorded in lobule V–VI and Crus I, supporting the link to zebrin-identity, rather than cortical location (*Figure 4B*). Moreover, the gradual trend that we observed for SS firing frequency, but not for CV2, in the different lobules in both the vermis and hemispheres was also observed for CS activity (*Figure 4C*, *Figure 3—figure supplement 1*). Since climbing fibers evoked prolonged EPSCs in Z+ Purkinje cells (*Paukert et al., 2010*), we also investigated the half-width of the first upward deflection in potential and the integrated deviation of the CS potential from zero. Both parameters were significantly higher in immunostaining identified Z+ PCs (half-width: *t* = −3.269, p=0.001, spike area: *t* = −2.523, p=0.013) (*Figure 4D*). Given that SS activity and the wave of CS activity correlated with zebrin, the distribution of post-CS configurations of SS activity might in principle also be affected (*Simpson et al., 1996*). Based on the peri-CS time histograms, we could distinguish four different types of SS responses

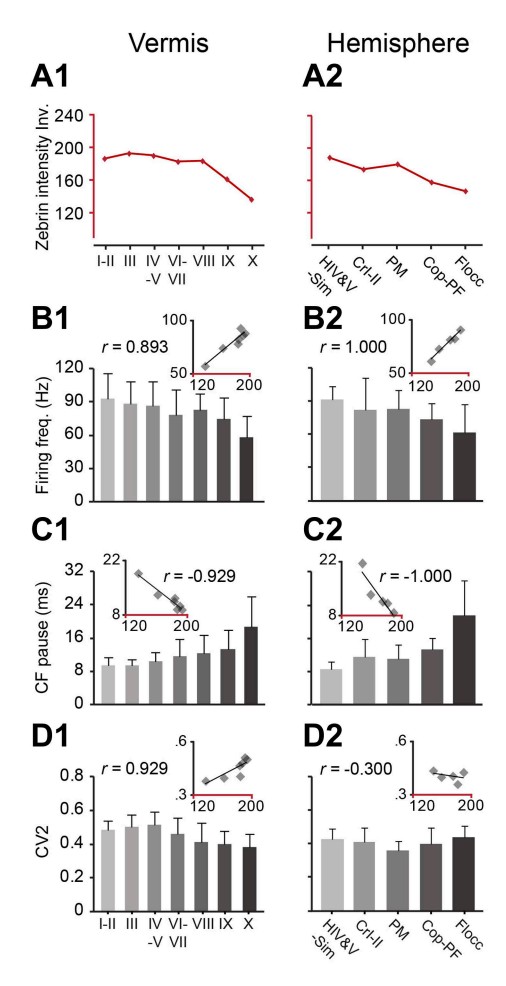

**Figure 3**. Zebrin staining intensity and simple spike frequency are inversely correlated. To test the correlation to zebrin identity of modules throughout the cerebellar cortex, Purkinje cell activity was recorded from all parts of the cerebellar cortex, each followed by dye injection to identify the lobule. (**A1–2**) The average zebrin staining intensities of Purkinje cell somata in vermis and hemispheres were obtained from the sagittal sections of three mice. Note that high intensity values equal weak staining, and vice versa. (**B1–2** and **C1–2**) The average simple spike firing frequency (vermis: n = 192 cells, 70 mice, r = 0.893, p=0.007; hemisphere: n = 53 cells, 30 mice, r = 1.000, p<0.001) and CF pause (vermis: r = −0.929, p=0.003; hemisphere: r = −1.000, p<0.001) show significant correlation with zebrin intensity over different parts of vermis and hemisphere. (**D1–2**) The CV2 of SSs could not be consistently related with zebrin intensity (vermis: r = 0.929, p=0.003; hemisphere: r = −0.300, p=0.624). Error bars represent SD. HIV&V-Sim, hemispheral part of lobule IV&V and simple lobule; CrI-II, Crus I and II; PM, paramedian lobule; Cop-PF, copula of the pyramis and parafloccu-lus; Flocc, flocculus.

*Figure 3. Continued on next page*

following the climbing fiber pause. These included a neutral pattern (i.e., normal type), a pattern with increased SS activity (i.e., facilitation type), and two patterns with decreased SS activity, one without and one with a superimposed oscillatory effect (i.e., suppression and oscillation type, respectively) (*Figure 4E*). Thus, if there is a relation with zebrin expression, one could predict that the facilitation type of cells prevail in the Z− zones, whereas the suppression and oscillation type of cells occur predominantly in the Z+ zones. This prediction did hold. Even though the normal type dominated in both Z− and Z+ PCs, the suppression and oscillation types only occurred in Z+ PCs. The facilitation type occurred in both Z− and Z+ PCs, but significantly more in the Z− areas (Z−: 17 /47, Z+: 6/57; $\chi^2$ = 9.835, p=0.002, Pearson's Chi-squared test) (*Figure 4F*). Attempts to find other parameters correlating with the response type were largely unsuccessful, except for the oscillation type, which showed a combination of low SS frequency and low CV (*Figure 4G*).

## Z+ and Z− Purkinje cells differ in intrinsic spiking activity

In general, SS activity of PCs results from the integration of their excitatory input, inhibitory input, and intrinsic pace-making activity (*De Zeeuw et al., 2011*). This raises the question as to what extent the difference in SS firing frequency between Z+ and Z− PCs results from differences in input or intrinsic activity. We used two approaches to tackle this question. First, we completely removed the impact of external inputs onto the PCs using blockers for AMPA, NMDA, and GABA$_A$ receptors during cell-attached recordings from sagittal slices (*Figure 5A*). The average SS firing frequency was, on average, 22 ± 7% lower over all lobules in vitro than that in vivo indicating that the larger part of SS activity is internally driven by PCs. The dominant impact of intrinsic PC activity was also reflected by the finding that the differential firing frequency pattern over all lobules in vitro correlated with that in vivo (r = 0.916, p=0.010, Pearson's correlation) (*Figure 5B*). For example, the in vitro SS firing frequency of PCs in lobules III, which are predominantly Z−, was alike the in vivo recordings significantly higher than that in lobule X, where PCs are Z+ (t = 2.844, p=0.007). This higher firing frequency in lobule III was associated with a higher intrinsic excitability in lobule III compared to lobule X, reinforcing the interpretation that the difference is predominantly intrinsic to Purkinje cells (*Figure 5—figure supplement 1*) (see also *Kim et al., 2012a*). To confirm that under these in vitro conditions, without excitatory or

*Figure 3. Continued*

The following figure supplements are available for figure 3:

**Figure supplement 1**. Statistical analysis of PC spiking characteristics per lobule.

inhibitory input, the difference in simple spike firing frequency still correlates with zebrin identity, we also recorded the activity of fluorescence-identified Z+ and Z− PCs in adjacent bands in slices from the EAAT4-eGFP mice. Comparison of sets from lobules II–V and lobule VIII–IX confirmed this presumption (II–V: $t$ = 2.910, p=0.017; VIII–IX: $t$ = 2.352, p=0.043) (*Figure 5C*).

To further assess the impact of excitatory and inhibitory inputs, we investigated the in vivo SS activity in mouse mutants, in which either the glutamatergic input (*a6-Cacna1a* mutants) or the GABAergic input (*PC-Δγ2* mutants) to PCs was affected. The *a6-Cacna1a* mutants are characterized by a silenced parallel fiber output in the vast majority of their granule cells due to a lack of voltage-gated calcium channels required for neurotransmission (*Galliano et al., 2013*), while the *PC-Δγ2* mutants are characterized by the absence of synaptic inhibition from the molecular layer interneurons through ablation of the γ2 subunit of the GABA$_A$-receptor in PCs (*Wulff et al., 2009*; *Figure 5D*). In both *a6-Cacna1a* and *PC-Δγ2* mutant mice the differences in vivo in SS firing frequency between lobules I–III and lobule X were still significant, analogous to that in normal mice (*a6-Cacna1a*, I-III: 75.1 ± 19.0 Hz, X: 50.2 ± 10.2 Hz, $t$ = 3.979, p<0.001; *PC-Δγ2*, I-III: 89.8 ± 14.9 Hz, X: 60.9 ± 15.6 Hz, $t$ = 4.876, p<0.001) (*Figure 5E*). In contrast, the CV2 values of SS activity were significantly reduced not only in vitro, but also in vivo in both *a6-Cacna1a* and *PC-Δγ2* mutants as compared to wild-types (*Figure 5F*). These differences held true for lobules I–III (in vitro: $t$ = 32.647; *a6-Cacna1a*: $t$ = 5.613, p<0.001; *PC-Δγ2*: $t$ = 3.068, p=0.003 vs in vivo wild-types), as well as for lobule X (in vitro: $t$ = 14.593, p<0.001; *a6-Cacna1a*: $t$ = 2.062, p=0.046; *PC-Δγ2*: $t$ = 3.292, p=0.002). Together, these data suggest that the SS firing frequency is largely determined by intrinsic properties of PCs, whereas the level of regularity appears to be predominantly determined by external inputs.

## Activation of TRPC3 contributes to increase in SS activity in Z− Purkinje cells

The finding that the difference in firing frequency between Z+ and Z− PCs must predominantly reflect their different intrinsic properties raises the question whether PC proteins other than zebrin also play a mechanistic role. This may be particularly relevant as zebrin, or aldolase C, is a glycolytic enzyme and probably plays a secondary role via energy consumption without a direct impact on the electrophysiological properties of PCs. In fact, when the products of aldolase C, that is glyceraldehyde-3-phosphate (GAP) and dihydroxyacetone phosphate (DHAP), were introduced to the ACSF in our in vitro recordings, SS firing increased in both the largely zebrin-negative lobule III and zebrin-positive lobule X (*Figure 6—figure supplement 1*), arguing against the possibility that aldolase C's enzymatic function directly contributes to a lower SS firing frequency in Z+ PCs. Hence, we shifted our focus to TRPC3, which can be associated with zebrin-negative PCs (*Mateos et al., 2001*; *Hartmann and et al, 2008*; *Kim et al., 2012b*), and underlies the mGluR1-mediated slow EPSCs (*Hartmann and et al, 2008*) and mGluR1-agonist (DHPG)-induced currents (*Nelson and Glitsch, 2012*), that have been shown to affect SS activity even in the absence of synaptic input (*Yamakawa and Hirano, 1999*; *Coesmans et al., 2003*; *Chanda and Xu-Friedman, 2011*). We first tested the effect of blocking TRPC3 on the activity of PCs in vitro in lobules III and X, in the absence of synaptic input, using two blockers, genistein and Pyr3 (*Kim et al., 2012b*; *Kiyonaka et al., 2009*). Both TRPC3 blockers had a significant impact on PC activity reducing the firing frequency in lobule III (genistein, p<0.001; Pyr3, p<0.001 vs vehicle control, one-way followed by Tukey's post-hoc test) without a significant effect in lobule X (p=0.271 and p=1.000 vs vehicle respectively, one-way ANOVA followed by Tukey's post-hoc test) (*Figure 6A,B*), an effect that is in line with that of blocking mGluR1 (*Yamakawa and Hirano, 1999*; *Coesmans et al., 2003*; *Chanda and Xu-Friedman, 2011*). To more directly compare the effect of blocking TRPC3 between lobule III and X, we recorded PC activity during wash-in of the blockers (*Figure 6C*). Wash-in of Pyr3 had a robust effect on SS firing frequency of PCs in lobule III (pre: 49.9 ± 7.9 Hz; post: 25.5 ± 9.9 Hz; $t$ = 5.412, p=0.002, paired Student's $t$ test) and this effect was significantly larger than that in lobule X (reduction, lobule III: 48.2 ± 19.7%; lobule X: 8.5 ± 16.6%; $t$ = 4.069, p=0.002) (*Figure 6D,E*).

An alternative candidate protein that has a zebrin-related expression in adult animals and could potentially influence spiking activity is EAAT4, a glutamate transporter that is expressed in a zebrin-like

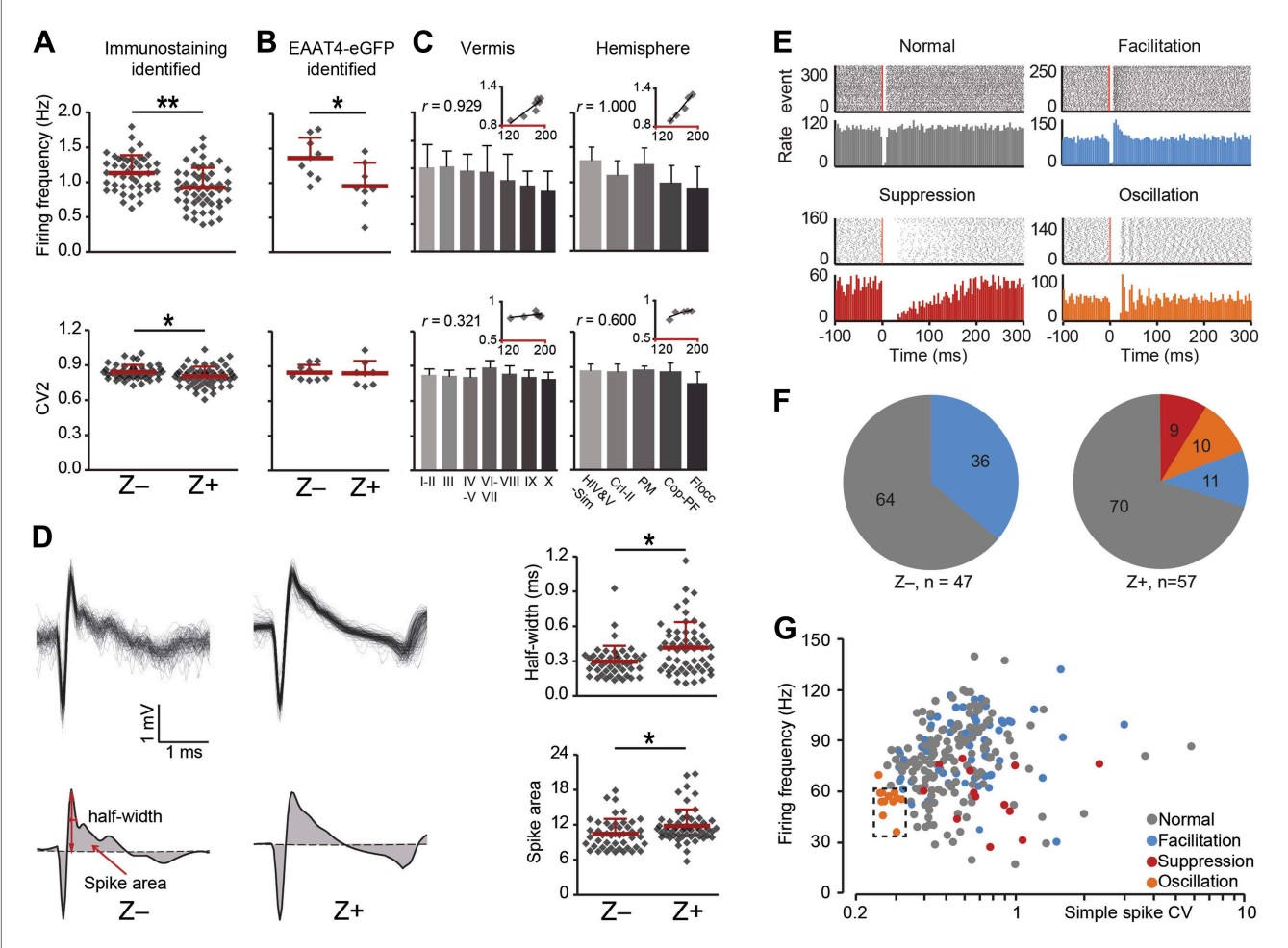

**Figure 4**. Complex spike characteristics depend on zebrin identitiy. (**A**) Similar to simple spike frequency, complex spike frequency differs between immunostaining identified Z+ and Z− PCs (data from zebrin-identified PCs shown in *Figure 1*; $t$ = 3.926, p<0.001), (**B**) This difference is confirmed in the sample of Z+ and Z− PCs obtained by two-photon imaging in EAAT4-eGFP mice, in that Z+ Purkinje cells have a lower complex spike firing frequency here too ($t$ = 2.692, p=0.017). **C**, Moreover, complex spikes frequency shows significant correlation with zebrin intensity in vermis and hemisphere (vermis: $r$ = 0.929, p=0.003; hemisphere: $r$ = 1.000, p<0.001). Even though the regularity of CSs differs between immunostaining identified Z− and Z+ PCs (**A**, bottom), this was not reproduced in the other two experimental data sets (**B**–**C**, bottom). (**D**) Typical Z− and Z+ CS shapes (−0.5 to +3 ms) showing the characteristics analyzed: half-width and spike area (left). Z+ PCs have a longer half-width and bigger spike area than Z− cells (right). (**E**) Raster plots of simple spike activity around complex spikes (event, −100 ms till +300 ms) were converted in peri complex-spike time histograms. Based on these histograms, we could distinguish four different types of simple spike response types among the Purkinje cells recorded in all areas: normal, facilitation, suppression and oscillation. (**F**) The percentage of different types in Z− and Z+ PCs (values indicate percentage). The facilitation type occurs predominantly in Z− PCs, whereas the suppression and oscillation type are restricted to the Z+ PCs. (**G**) Attempts to find other parameters correlating in all recorded cells (n = 243 cells) with the response type were largely unsuccessful. The exception is the oscillation type, which has a signature combination of simple spike frequency and CV (11 out of 13, SS freq. range 35–60 Hz and CV <0.32). Two-photon imaging data are only included in panel **B**; panels **D**–**F** are based on immunostaining identified Z+ and Z− PCs only and panels **C** and **G** on all recorded PCs. Error bars represent SD, *p<0.05, **p<0.001.

pattern and carries a depolarizing current (*Wadiche and Jahr, 2005*). Based on the higher expression of EAAT4 in Z+ Purkinje cells, blocking EAAT4 would arguably affect lobule X more than lobule III, but the general EAAT blocker DL-TBOA had no significant effect on the activity of Purkinje cells in either lobule (*Figure 6—figure supplement 2A,B*) (lobule III: $t$ = 1.219, p=0.227; lobule X: $t$ = −0.597, p=0.533), maintaining the difference between lobule III and X (inset, $t$ = 3.641, p<0.001). Wash-in of DL-TBOA did also not affect activity in lobule III or X (lobule III: 2.0 ± 10.8%; lobule X: −3.6 ± 3.4%; $t$ = 0.982, p=0.352) (*Figure 6—figure supplement 2C–E*).

Next, we studied whether the effects of TRPC3 blockers were sufficiently robust to also induce measurable effects in vivo. In line with the in vitro data, both genistein and Pyr3 (i.p. and i.c.v., 240 mg/kg

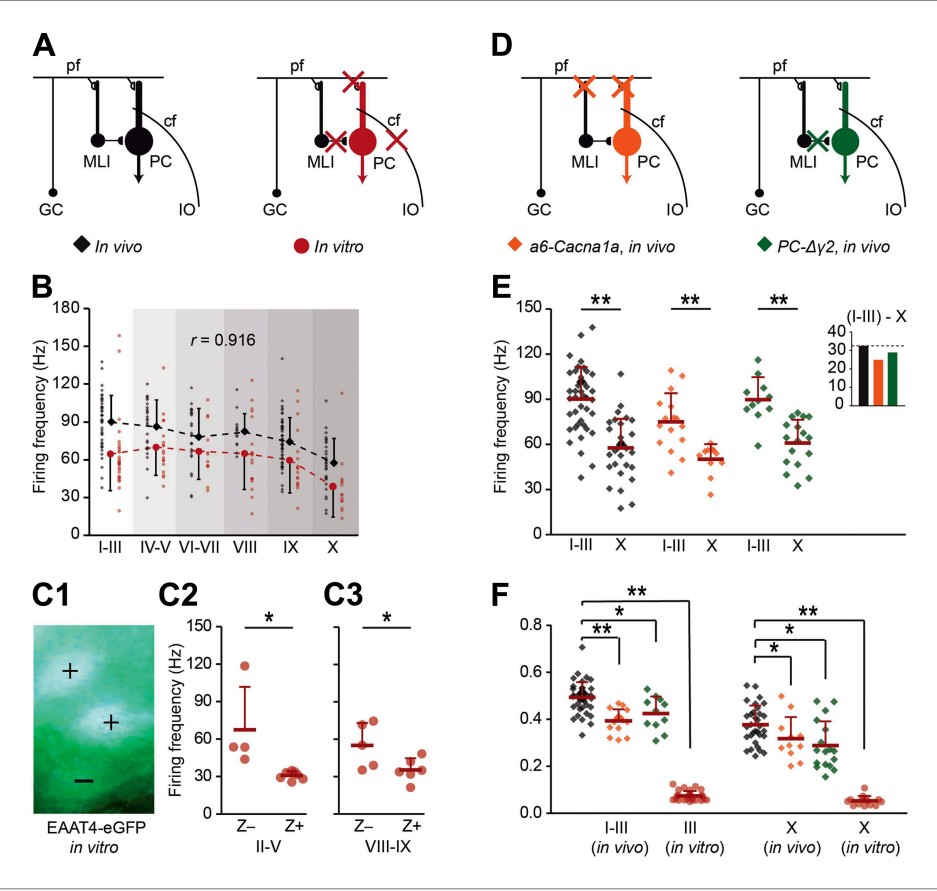

**Figure 5**. Zebrin-related differences are present in the intrinsic activity of Purkinje cells. To test if intrinsic or input-related differences underlie the difference in simple spike frequency, we recorded PC activity in conditions of limited or no synaptic input. (**A**) PC activity was recorded in vitro (n = 107 cells, 15 mice) under complete block of synaptic inputs. (**B**) Spiking frequency in vitro (red) was lower than that in vivo (black) over the range of lobules, but the shape of the curve was similar (r = 0.916, p=0.010, Pearson's correlation). (**C1–3**) To verify the correlation with zebrin, we recorded activity of EAAT4/zebrin-positive and negative PCs in slices of EAAT4-eGFP mice. Both in lobules II–V (Z+: n = 7 cells, Z−: n = 4; 3 mice; t = 2.910, p=0.017) and lobules VIII-IX (Z+: n = 6, Z−: n = 5; 2 mice; t = 2.352, p=0.043) the difference in simple spike firing frequency was present, further confirming the link with zebrin. (**D**) Next, extracellular recordings were made in vivo in a6-Cacna1a and PC-Δγ2 mutant mice that have minimized excitatory and no synaptic inhibitory inputs to their PCs, respectively. (**E**) PC activity in Z+ lobule X of both mutants was lower than that in the predominantly Z− lobules I–III (wild types, lobules I–III: n = 43 cells, 18 mice, lobule X: n = 32 cells, 25 mice, t = 6.808, p<0.001; a6-Cacna1a, I–III: n = 16 cells, 2 mice; X: n = 11 cells, 2 mice; t = 3.979, p<0.001; PC-Δγ2, I–III: n = 11 cells, 3 mice; X: n = 17 cells, 3 mice; t = 4.876, p<0.001). Inset compares the absolute differences in firing frequency between lobules I–III and X. (**F**) CV2 values of Z− and Z+ SS activity from in vitro recordings (lobules I–III and X: both p<0.001) and in vivo recordings of both a6-Cacna1a mutants (lobule I–III: t = 5.613, p<0.001; lobule X: t = 2.062, p=0.046) and PC-Δγ2 mutants (lobules I–III and X: both p<0.005) were significantly lower than the wild type recordings. Abbreviations: cf, climbing fiber; GC, granule cell; IO, inferior olive; MLI, molecular layer interneuron; PC, Purkinje cell; pf, parallel fiber. Error bars represent SD, *p<0.05, **p<0.001.

The following figure supplements are available for figure 5:

**Figure supplement 1**. Purkinje cell intrinsic excitability is higher in lobule III than in X.

and 200 µg, respectively) caused a decrease in SS activity in lobules I–III, lasting for several hours (vehicle: 90.6 ± 13.3 Hz; genistein: 74.6 ± 14.9 Hz; Pyr3: 61.6 ± 15.5 Hz; both p<0.001 vs vehicle, One-way ANOVA followed by Tukey's post-hoc test), while no significant effect was recorded in lobule X (vehicle: 52.9 ± 13.3 Hz; genistein: 57.5 ± 20.8 Hz,; Pyr3: 51.1 ± 14.3 Hz, p=0.546 and p=0.887 vs vehicle, One-way ANOVA followed by Tukey's post-hoc test) (***Figure 6F,G***). Together with changes in

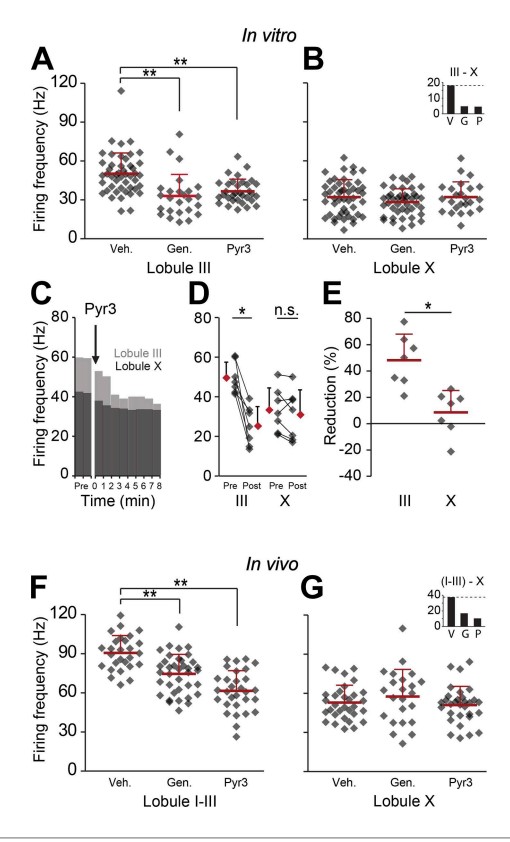

**Figure 6**. Blocking TRPC3 attenuates the simple spike frequency difference. In search for the underlying mechanism, we tested the contribution of TRPC3, which can be indirectly linked to zebrin-like expression. (**A** and **B**) The presence of TRPC3 blocker genistein (10 μM) or the more selective Pyr3 (100 μM) reduced Purkinje cell firing frequency in lobule III (vehicle: n = 47 cells, 6 mice; genistein: n = 25 cells, 7 mice; Pyr3: n = 33 cells, 7 mice; both p<0.001 vs vehicle, One-Way ANOVA followed by Tukey's post-hoc test), but not in lobule X (vehicle: n = 48 cells, 6 mice; genistein: n = 44 cells, 7 mice; Pyr3, n = 24 cells, 7 mice; p=0.271 and p=1.000 vs vehicle, respectively, One-Way ANOVA followed by Tukey's post-hoc test), virtually eliminating the difference between averages for lobule III–X (inset). To more directly quantify the effect of blocking TRPC3, we washed-in Pyr3 during the recording of Purkinje cells in lobule III and X. (**C**–**E**) Pyr3 wash-in significantly decreased the simple spike firing frequency in lobule III (n = 7 cells, 7 mice; t = 5.412, p=0.002, paired Student's t test), and this decrease was larger in lobule III than in lobule X (t = 4.069; p=0.002). (**F**–**G**) In line with the in vitro data, in vivo blocking of TRPC3 by application of genistein (240 mg/kg, i.p.) or Pyr3 (200 μg, i.c.v.) decreased simple spike firing in lobule I–III (vehicle: n = 27 cells, 3 mice; genistein: n = 37 cells, 3 mice and Pyr3: n = 30 cells, 2 mice; both p<0.001 vs vehicle, one-way ANOVA followed by Tukey's post-hoc test), but had no effect in lobule X (vehicle: n = 32 cells, 3 mice; genistein: n = 23

*Figure 6. Continued on next page*

simple spike frequency, several other parameters, related to the complex spike of Purkinje cells in lobule I-III shifted, upon Pyr3 application, towards the values for lobule X with or without drugs. These included including climbing fiber pause, complex spike frequency and width and type of simple spike response following complex spikes (*Figure 6—figure supplement 3*). Effects of genistein were less consistent, probably due to its less selective nature. Together, these data suggest that TRPC3 contributes to the elevated SS activity in zebrin-negative PC zones.

## Discussion

This study provides, to our knowledge, the first evidence for ubiquitously organized differences in cerebellar Purkinje cell firing activity in vivo and for the correlation of these differences to a marker protein, zebrin II. As zebrin, as a biomarker, can be associated with the modular organization of the olivo-cortico-nuclear circuit (*Brochu et al., 1990*; *Voogd et al., 2003*; *Pijpers et al., 2006*; *Sugihara and Shinoda, 2007*), our results indicate that Purkinje cells within the same module operate around a preferred range of intrinsically determined SS firing frequencies and that this activity is different from modules with a different zebrin identity. Moreover, presumably as a secondary effect, CS activity is also altered. Since the differences were consistently found throughout the entire cerebellar cortex, these findings have direct consequences for all cerebellar functions and the coding schemes they can engage.

### Difference in SS activity and underlying factors

The difference in SS firing frequency between zebrin-positive and zebrin-negative PCs in vivo was robust (i.e., approximately 60 Hz vs 90 Hz) and highly significant, present throughout the cerebellar cortex and could be reproduced by directly comparing the activity of Purkinje cells in adjacent modules. Although we cannot exclude the possibility that lobule-specific effects contribute to the observed differences in simple and complex spike firing frequency, the results obtained in EAAT4-eGFP mice in vivo and in vitro argue against a contribution of their rostro-caudal or lobular location. The difference in SS firing frequency is probably largely determined by the intrinsic properties of PCs, as this difference was maintained in the reduced slice preparation, in which the inputs are blocked, as well as in mouse mutants, in which the excitatory (*a6-Cacna1a* mice) or inhibitory (*PC-Δγ2* mice) inputs are attenuated. Comparing the PC activity in zebrin-positive and zebrin-negative PCs

*Figure 6. Continued*

cells, 4 mice and Pyr3: n = 31 cells, 4 mice; p=0.546 and p=0.887 vs vehicle, respectively one-way ANOVA followed by Tukey's post-hoc test), resulting in a pronounced reduction of the difference (inset). Error bars represent s.d., *p<0.05, **p<0.001.

The following figure supplements are available for figure 6:

**Figure supplement 1**. Aldolase C enzymatic reaction products GAP and DHAP increase the activity in lobules III and X.

**Figure supplement 2**. Effects of blocking EAAT4 on Purkinje cell activity in lobule III and X in vitro.

**Figure supplement 3**. Effects of TRPC3 blockers on other PC activity parameters.

per transverse zone confirmed its link to zebrin-identity, but also revealed a more subtle difference within the population of zebrin-positive PCs. These differences could be lobule-specific and/or originate from differences in input, in more subtle variations in zebrin or its related proteins or even in the expression pattern of other proteins. It should be noted, however, that blocking TRPC3 shifts the activity of Z− PCs towards that of Z+ PCs, indicating that the potential differences within the group of Z+ PCs do not affect our conclusions.

In contrast, the level of regularity of firing (i.e., CV2) was not consistently dependent on zebrin identity, but significantly altered by impairing the excitatory and/or inhibitory inputs. In line with the notion that reduced SS activity of PCs, as observed in the zebrin-positive modules, should lead to enhanced firing of the GABAergic neurons in the cerebellar nuclei and thereby to reduced activity in the inferior olivary neurons (*Chen et al., 2010*; *De Zeeuw et al., 2011*), we found that CS activity induced by activity of olivary climbing fibers was reduced in zebrin-positive PCs. In fact, reduction of simple spike frequency in lobule III in vivo for several minutes to hours by Pyr3 application also reduced complex spike frequency, supporting the indirect control of simple spikes on complex spike activity. Interestingly, temporary increases in climbing fiber-evoked CS activity suppress SS frequency providing a homeostatic control mechanism within an olivocerebellar module (*Mathews et al., 2012*; *Coddington et al., 2013*). Thus, whereas the external inputs to PCs may control the precise temporal coding of SS activity at rest as well as the firing frequency and dynamic range during natural sensory stimulation (*Badura et al., 2013*; *Galliano et al., 2013*), their intrinsic properties appear to determine the baseline frequencies of SSs as well as CSs at rest, around which they can operate.

These findings raise the question which proteins in the zebrin-positive and zebrin-negative zones may actually determine the difference in intrinsic firing frequencies of their PCs. Since zebrin's enzymatic reaction products did not underlay the differences in SS firing frequency, we shifted our attention to other proteins that are expressed in pattern similar or complementary to that of zebrin. We targeted EAAT4 that is expressed in a pattern similar to that of zebrin, and TRPC3, the effector channel of a cascade of proteins that has zebrin-like expression patterns (*Dehnes et al., 1998*; *Mateos et al., 2001*; *Wadiche and Jahr, 2005*). Although blocking EAAT4 in vitro had no detectable effect on SS firing frequency, blocking TRPC3 reduced SS activity in lobule III (largely zebrin-negative), but not in lobule X (zebrin-positive), both in vitro and in vivo. It should be noted that, although TRPC3 gene expression appears to vary from anterior to posterior with higher levels in the anterior, largely zebrin-negative, lobules (Allen Brain Atlas, www.brain-map.org), there is no immunohistochemical evidence for differences in protein levels (*Hartmann and et al, 2008*; *Becker et al., 2009*; *Sekerkova et al., 2013*). If the expression of TRPC3 is indeed homogeneous throughout the cerebellum, the differential effect of blocking TRPC3 suggests that its activity might be higher in Z− PCs. Two mutually non-exclusive mechanisms could contribute to this difference in activity. First, several proteins in the molecular cascade related to TRPC3 are expressed in zebrin-like bands, including the IP3-receptor (*Furutama et al., 2010*) (TRPC3 modulator [*Kim et al., 2012b*]), PLCβ3/4 [*Sarna et al., 2006*] (TRPC3 activator [*Kim et al., 2012b*]), PKCδ [*Barmack et al., 2000*], and NCS-1 [*Jinno et al., 2003*]. In fact, zebrin II or aldolase C, which is not likely to be involved via its enzymatic function, bears the capacity to bind IP3 (*Baron et al., 1999*), and thus could potentially reduce the activation of TRPC3 in Z+ PCs, through the IP3-receptor. At the same time, mGluR1 subtype b is expressed in a pattern complementary to that of zebrin (*Mateos et al., 2001*), and it has been shown that mGluR1 can be tonically activated, that mGlurR1 blockers can reduce SS firing frequency (*Yamakawa and Hirano, 1999*; *Coesmans et al., 2003*; *Chanda and Xu-Friedman, 2011*), and that mGluR1-evoked depolarizing currents can be blocked with TRPC3-selective blockers (*Kiyonaka et al., 2009*; *Kim et al., 2012b*). However, the possibility that an alternative pathway, independent of mGluR1, leads to TRPC3 activation cannot be

excluded. Knowledge of this pathway of proteins and their exact interactions is at current presumably incomplete and beyond the scope of this study. The findings that the expression patterns of mGluR1b, PLCβ, PKCδ, and IP3R1, all of which are key proteins in calcium release from intracellular calcium stores, are linked to cerebellar modules (*Mateos et al., 2001*) and intimately connected with TRPC3, provokes the speculation that this entire pathway contributes to the difference in SS activity between zebrin-positive and zebrin-negative PCs (*Hartmann and et al, 2008*; *Becker et al., 2009*).

## General functional implications

Our finding that SS activity and indirectly also CS activity at rest are determined by the intrinsic properties of PCs implies that they operate around these baseline frequencies during natural stimulation and behaviour. Interestingly, the low and high baseline frequencies of zebrin-positive and zebrin-negative PCs also appear to be in line with their propensities for induction of long-term potentiation (LTP) and long-term depression (LTD), respectively (*Wadiche and Jahr, 2005*; *Wang et al., 2011*). Thus, PCs operating at lower frequencies may be preferentially potentiated, whereas PCs with higher SS firing frequency may have less 'space' for increasing the firing rate and may be more prone to express LTD. Likewise, entrainment of cerebellar nuclei neurons by synchronized SS input from PCs, which results in phase-locking of connected neurons, may occur at 50–80 Hz, but is impaired at 100 Hz (*Person and Raman, 2012*). If correct, this mechanism predicts that the phase-locking mechanism is engaged in contacts between zebrin-positive PCs and cerebellar nuclei neurons, whereas those involved in zebrin-negative zones may be more prone for rebound excitation, which follows strong forms of inhibition (*De Zeeuw et al., 2011*; *Person and Raman, 2012*).

The cerebellar nuclei can also be divided based on the zebrin expression pattern, with a rostral half that receives, predominantly, input from zebrin-negative PC's and a caudal part that mostly receives zebrin-positive inputs (*Sugihara and Shinoda, 2007*; *Sugihara et al., 2009*; *Sugihara, 2011*). This implies that PC input to cerebellar nuclei neurons may be segregated on the basis of frequency, and that as a consequence the output of cerebellar nuclei neurons located within zebrin-positive and zebrin-negative territories may be distinctly different. Although this remains speculative at this stage, similar phenomena have been described for highly active neurons in cerebral cortex (*Yassin et al., 2010*) and hyperpolarization-activated currents in affiliated olfactory bulb mitral neurons (*Angelo et al., 2012*). Combined with the zebrin- or lobule-related prevalence of plasticity mechanisms (*Wadiche and Jahr, 2005*; *Wang et al., 2011*), our results suggest that the biochemically identified bands in the structurally homogenous cerebellar cortex are physiologically different with distinct biophysical signatures that probably have significant implications downstream in the cerebellar nuclei and thereby on motor behaviour and cognition.

## Materials and methods

### In vivo extracellular recordings

We recorded in vivo single-unit Purkinje cell activity in adult male C57Bl/6 mice (C57Bl/6J, Charles River), aged 10–35 weeks. Mice were prepared for recordings by placing an immobilizing construct (pedestal) and a craniotomy on their skulls (*Badura et al., 2013*). In short, the skin over the skull was shaven, and opened along the rostro-caudal midline. Using Optibond (Kerr, Salerno, Italy) and Charisma (Heraeus Kulzer, Hesse, Germany), a U-shaped holder (6 × 4 mm) with a magnet inside (4 × 4 mm, MTG, Weilbach, Germany) was fixed on the skull, overlying the frontal and parietal bones. Next, the medial neck muscles overlying the occipital bone were removed, a craniotomy was made over the interparietal or occipital bone and a recording chamber was placed around it, allowing in vivo electrophysiological recordings throughout different areas in the cerebellum of awake mice. The exact location of the craniotomy depended on the target area, see *Figure 1—figure supplement 1* for details. After recovery of >24 hr, mice were head-fixed to a bar, their bodies restrained in a custom-made plastic tube and the dura was opened to facilitate the recording of extracellular Purkinje cell activity, as previously described (*Hoebeek et al., 2005*). Electrophysiological activity in the cerebellar cortex was recorded using double barrel borosilicate glass pipettes (theta septum, 1.5 OD, 1.02 ID, WPI, FL, USA). To do so, one of the barrels was opened laterally, approximately 10 mm from the taper, to allow entrance of the electrode wire and sealed with glass glue at the back. The other barrel was filled with a blue dye (Alcian Blue, 0.1–0.2% solution in saline; Sigma–Aldrich, St. Louis, MO, USA). The recording half of the double barrel pipettes were filled with 2 M NaCl-solution,

and had a tip size of 3–6 µm, respectively. Pipettes were advanced into the cerebellum with an oil micro-drive (Narishige, Tokyo, Japan) and signals were pre-amplified (custom-made preamplifier, 1000x DC), filtered (CyberAmp 320, Axon, Molecular Devices, Sunnyvale, CA, USA), digitized (Power1401, CED, Cambridge, UK), and stored for offline analysis. After successful recordings, brief pressure pulses were delivered through the other barrel of the electrode, using a custom-built device, to mark the recording site. To obtain *a6-Cacna1a* and *PC-Δγ2* mice, we used the Cre/loxP system to delete exon 4 of the gene coding for the P/Q-type voltage-gated calcium channel (*Cacna1a*) selectively from granule cells and exon 4 of the GABA$_A$ receptor γ2 subunit gene (*Gabrg2*) selectively from PCs, respectively, as described previously (*Wulff et al., 2009*; *Galliano et al., 2013*). In short, we crossed *Cacna1a$^{lox/lox}$* mice with Gabra6::Cre (or Δa6::cre) mice (*Aller et al., 2003*) and *Gabrg2$^{lox/lox}$* mice with Pcp2::Cre (or L7::Cre) mice (*Oberdick et al., 1990*), respectively. From the offspring, that was heterozygous for the floxed genes (i.e., *Cacna1a$^{lox/+}$* and *Gabrg2$^{lox/+}$*), Cre-negative males were crossed with Cre-positive females to generate, amongst others, Δa6::cre;Cacna1a$^{lox/lox}$ (or *Cacna1a$^{Δ/Δ}$*, here named *a6-Cacna1a*) and Pcp2::cre;gabrg2$^{lox/lox}$ (or *Gabrg2$^{Δ/Δ}$*, here named *PC-Δγ2*) mice, respectively. Both lines were maintained in a C57Bl6 background. In the experiments with mutant mice and blocker injections, double and single barrel (2.0 mm OD, 1.16 mm ID, Harvard Apparatus, MA, USA) borosilicate glass pipettes were used, and alcian blue was injected to confirm that the recordings were from lobules I–III or X.

## In vivo two-photon imaging of EAAT4-eGFP mice

EAAT4-eGFP mice express enhanced green fluorescent protein (eGFP) under control of the EAAT4 promoter, and were generated using the bacterial artificial chromosome (BAC) (*Gincel et al., 2007*). Targeted recordings of eGFP-positive and eGFP-negative Purkinje cells were made after visualizing the eGFP-positive bands using in vivo two-photon imaging of 5 awake EAAT4-eGFP mice (3 females, 2 males, 10–26 weeks old). Images were acquired using a TriM Scope II (LaVision BioTec, Bielefeld, Germany) attached to an upright microscope with a 40x/0.8 NA water-immersion objective (Olympus, Tokyo, Japan). Laser illumination was provided by a Chameleon Ultra titanium sapphire laser (Coherent, Santa Clara, CA). We aimed to image Purkinje cells in the superficial layer of a restricted part of the cortex (lobules V-VI and Crus I) at a depth of ~250 µm using an excitation wavelength of 920 nm, and their location in relation to zebrin bands was determined online. The recording pipette was filled with Alexa-594 (10 µM in 2 M NaCl; Life Technologies, Carlsbad, CA) and visualized with an excitation wavelength of 800 nm, the minimum recording duration was 30 s. Images from eGFP and Alexa-594 were filtered using a Gaussian kernel, contrast-optimized and subsequently merged in Photoshop (Adobe, San Jose, CA). Purkinje cells recorded in vivo from EAAT4-eGFP mice are included in *Figure 2*, *Figure 2—figure supplement 1*, *Figure 4B*.

## Analysis of in vivo recordings

Purkinje cells were identified by the occurrence of simple and complex spikes and were confirmed to be from a single unit by the presence of a pause in simple spikes after each complex spike. To assure the quality and reliability of the recording the following criteria were imposed: (1) a minimum recording duration of 120 s, (2) stable simple spike amplitude, (3) no clear signs of tissue damage in a circle with 400 µm radius, around the recording site (*Figure 1—figure supplement 1B*). All in vivo data were analyzed using SpikeTrain (Neurasmus BV, Rotterdam, The Netherlands, www.neurasmus.com), running under Matlab (Mathworks, MA, USA). SpikeTrain uses wave clustering to identify simple and complex spikes, and in case of doubt manual checking (and correcting) would be performed. For each cell the firing rate, CV and mean CV2 were determined for simple and complex spikes, as well as the climbing fiber pause. CV is the standard deviation of inter-spike intervals (ISI) divided by the mean, the mean CV2 is calculated as the mean of $2 \cdot |(ISI)_{n+1} - ISI_n| / (ISI_{n+1} + ISI_n)$. Both are measures for the regularity of the firing, with CV reflecting that of the entire recording and mean CV2 that of adjacent intervals, making the latter a measure of regularity on small timescales. The climbing fiber pause is determined as the minimum duration between a complex spike and the following simple spike. To extend this analysis, we also plotted histograms of simple spike activity time locked on the complex spike, and labelled the shape of this time histogram as normal, facilitation, suppression, and oscillation (see *Figure 3* for examples). The spike characteristics half maximum width (HMW) and spike area were determined from the normalized average signal of simple and complex spikes of individual recordings. Half-width was calculated as the width of the first peak at half of its maximum amplitude. The spike

area was defined as the integral of the rectified complex spike wave form in a time window of 0.5 ms pre and 3 ms post spike onset.

## In vitro cell-attached and whole-cell patch recordings

Acute sagittal slices (250 µm thick) were prepared from the cerebellar vermis of 3–5 month old male C57BL/6J mice (Charles River) in ice-cold slicing medium that contains the following (in mM): 240 sucrose, 2.5 KCl, 1.25 $Na_2HPO_4$, 2 $MgSO_4$, 1 $CaCl_2$, 26 $NaHCO_3$, and 10 D-glucose, bubbled with 95% $O_2$ and 5% $CO_2$. Subsequently, slices were incubated in ACSF containing (in mM): 124 NaCl, 2.5 KCl, 1.25 $Na_2HPO_4$, 2 $MgSO_4$, 2 $CaCl_2$, 26 $NaHCO_3$, and 10 D-glucose equilibrated with 95% $O_2$ and 5% $CO_2$ at 34.0°C for 30 min, and then at room temperature. Slices were typically used within 5 hr ex vivo. NBQX (10 µM), DL-AP5 (50 µM), and picrotoxin (100 µM) were bath-applied to block AMPA-, NMDA-, and GABA subtype A ($GABA_A$)-receptors, respectively. Borosilicate glass pipettes (WPI) were filled with ACSF and had an open pipette resistance of 2–4 MΩ. Purkinje cells were identified using visual guidance by DIC video microscopy and water-immersion 40X objective (Axioskop 2 FS plus; Carl Zeiss, Jena, Germany). Slices were transferred to the recording chamber and incubated for at least 10 min before starting the recordings. We recorded the Purkinje cell activity in cell-attached mode (0 pA injection) at 33.0 ± 1.0°C, with a distance of 0.5 cm between temperature probe and slice. Current clamp recordings were performed with the same setting as cell-attached recording, except that pipettes were filled with intracellular solution contains the following (in mM): 120 K-Gluconate, 9 KCl, 10 KOH, 3.48 $MgCl_2$, 4 NaCl, 10 HEPES, 4 $Na_2ATP$, 0.4 $Na_3GTP$, and 17.5 sucrose, pH 7.25 and Osm 295.

For experiments in EAAT4-eGFP mice, all experimental conditions were the same as cell-attached experiments above, except that coronal slices (300 µm thick) were used to record from identified EAAT4-positive and EAAT4-negative Purkinje cells within the same lobules. We first identified the lobules by their locations and band patterns with a 10X objective, and then zoomed in with a 40X objective to proceed to recording.

## In vitro data acquisition and analysis

Electrophysiological data were acquired using an EPC9 amplifier (HEKA, Lambrecht, Germany), filtered at 10 kHz and digitized at 25 kHz. Acquisition was controlled using PULSE software (HEKA) and the data were exported and analyzed using Minianalysis (v6.0.3) software (Synaptosoft, Fort Lee, NJ, USA) or Matlab. The typical signal-to-noise ratio was larger than 5:1, and minimum recording duration was 120 s, unless stated otherwise. Cells were included based on the following criteria: (1) the CV over the whole period of recording was <0.2; (2) the average frequency changed less than 20% between the first and the last (30 s), except for those in the wash-in experiment. To minimize the day-to-day and slice-to-slice variations, recordings were targeted at different lobules for every slice. For the Pyr3 wash-in experiment, firing frequencies were normalized to the average frequency over the 2-min period before adding the drug to the ACSF (pre). The wash-in effect was determined by calculating the firing frequency in the period of 5–7 min after the drug was in the recording chamber (post).

In the whole-cell patch recording, the membrane potential of Purkinje was held at −65 mV using current injection to avoid spontaneous spiking activity (average: −454 ± 38 pA). We recorded the intrinsic excitability by injecting depolarizing currents ranging from 100 to 1000 pA (100 pA steps) relative to the holding current. Data were exported and analysed using threshold search with Clampfit (v10.4, Molecular Devices, Sunnyvale, CA, USA).

## Drugs

DL-TBOA (EAAT blocker, Tocris, Ellisville, MO, USA), genistein (TRPC3 blocker, Sigma–Aldrich), and Pyr3 (TRPC3 blocker, Tocris) were dissolved in dimethyl sulfoxide (DMSO, Carl Roth GmbH, Karlsruhe, Germany) and ACSF. For in vivo recordings, mice were injected with 240 mg/kg genistein (i.p.) and 200 µg Pyr3 (i.c.v) dissolved in saline or DMSO. Vehicle control mice were injected with 100 µl DMSO (i.p.). After injection, extracellular Purkinje cell activity was recorded as described above, and alcian blue was injected to verify that the recordings were from lobules I-III or X. The minimum recording duration was 60 s and recordings were made for up to 4 hr after injection for these experiments. For in vitro experiments all blockers were prepared in 1:1000 stock solutions in DMSO, stored at −20°C and used within 4 weeks after preparation. Blockers were bath applied where indicated in the following concentrations: DL-TBOA (25 µM), genistein (10 µM), and Pyr3 (100 µM). Except for the wash-in

experiments, all recordings started after the slice was incubated in the drug-containing ASCF for at least 15 min. In the cases where we observed an effect in the wash-in experiments, the firing frequency of Purkinje cells typically dropped the moment the drug reached the bath, an effect maximizing within a few minutes and sometimes followed by a smaller recovery. The experimenter was blind to the presence and type of drug applied until analysis was completed. The tubing was changed after every blocker experiment.

## Histochemistry

After recordings, mice were deeply anesthetized with Nembutal and perfused with 75 ml of 4% paraformaldehyde (PFA). The brains were removed from the skull and post-fixed for 1–2 hr in 4% PFA, and stored in 0.1M PB containing 10% sucrose. After embedding in 10% gelatin and 10% sucrose, blocks were hardened in a solution containing 10% formaldehyde, 30% sucrose for 1–2 hr at room temperature and then stored overnight in 0.1M PB with 30% sucrose at 4°C. To identify if recordings were made in a Z+ or Z− band, coronal sections with a thickness of 40 μm were processed based on a standard immunostaining procedure; next, they were thoroughly rinsed with 0.1MPB. The goat-derived zebrin II antibody (Santa cruz, TX, USA) was diluted at 1:1000 in PBS, pH 7.6, containing 2% normal horse serum and 0.4% Triton. Rabbit anti-goat secondary antibody HRP conjugate diluted at 1:200 was used as a secondary antibody (Dako, Glostrup, Denmark). The sections were thoroughly rinsed three times with PBS and PB, followed by diaminobenzidine (DAB) incubation (0.66% DAB, and 0.033% $H_2O_2$ for 10–20 min). The sections were put on glass and then dehydrated by different grades of ethanol (70%, 80%, 90%, 96%, 96%, 100%, 100%, 100%, 2 min per grade), xylene was applied to clean the ethanol, and subsequently the sections were covered with Permount. To only determine the recording sites for recordings throughout the cerebellum, sagittal sections were cut at 80 μm followed by neutral red staining.

## Analysis of immunohistochemistry

The injection sites were located by the light microscope. If the same injection could be found in several slices, the injection was allocated to the slice with the highest density of Alcian Blue. If the injection site was at the border of two cerebellar lobules, the cell was allocated to the most rostral lobule. Particularly in the hemipsheres, a distinction between positive and lightly positive areas can be made. In this study, they were taken together as positive, and compared to the—clearly identifiable—negative bands. Recordings were excluded from further analysis if there was clear tissue damage within a circle with a radius of 400 μm around the injection site. The recording sites of 7 out 8 cells in the flocculus (FL) were confirmed by the response to visual stimulation instead of histology (*Galliano et al., 2013*). Example sections and sections used to determine the staining intensity of zebrin II were photographed using a Leica DMRB microscope equipped with Leica DC300 camera. To compare zebrin intensity between lobules the average pixel intensity of the PC somas in each lobule was determined using ImageJ software, based on the average of a total of 3 sections for vermis and 2 sections for the hemisphere per mouse, from 3 mice. Sections compared in the same panel, were processed in parallel. To correct for minor differences in overall staining intensity between vermis and hemispheres, soma intensities were normalized based on background (granule cell layer) intensity.

## Statistics

All values are shown as mean ± SD, unless stated otherwise. Unpaired Student's *t* test were used for comparisons and Spearman's *r* test for correlations, unless stated otherwise, and p<0.05 was considered to be significant. Comparisons in which at least one of the groups had n ≤ 4 were re-tested using a Mann–Whitney *U*-test, and in all cases the outcome was confirmed.

All experiments were performed under the GGO license no. IG 04-197, and approved by the Dutch animal ethical committee (DEC, EMC 2168/2545/2999/3002/3057).

## Acknowledgements

The authors want to thank Mandy Rutteman, Erika Goedknegt, Elize Haasdijk, Laura Post, and Daphne Groeneveld for technical assistance and Gerard Borst and Jan Voogd for valuable discussion. We thank Jeremy Rothstein for providing the EAAT4-eGFP mice.

# Additional information

## Funding

| Funder | Author |
| --- | --- |
| Erasmus University Rotterdam Fellowship | Freek E Hoebeek, Martijn Schonewille |
| Dutch Organization for Medical Sciences | Chris I De Zeeuw |
| Dutch Organization for Life Sciences | Freek E Hoebeek, Chris I De Zeeuw, Martijn Schonewille |
| European Community | Chris I De Zeeuw |
| Prinses Beatrix Fonds | Chris I De Zeeuw |
| China Scholarship Counsil | Chiheng Ju |

The funders had no role in study design, data collection and interpretation, or the decision to submit the work for publication.

## Author contributions

HZ, Designed, performed and analyzed the in vivo electrophysiology experiments, Designed, performed and analyzed the two-photon experiments, Designed, performed and analyzed the immunohistochemistry, Wrote the manuscript; ZL, Designed, performed and analyzed the in vitro electrophysiology experiments, Wrote the manuscript; KV, Wrote the software and guided the analysis of in vivo electrophysiology experiments, Designed, performed and analyzed the two-photon experiments, Contributed to manuscript preparation; CJ, Designed, performed and analyzed the two-photon experiments, Contributed to manuscript preparation; ZG, Designed, performed and analyzed the in vitro electrophysiology experiments, Contributed to manuscript preparation; LWJB, Designed, performed and analyzed the two-photon experiments, Contributed to manuscript preparation; TJHR, Guided the project, Contributed to manuscript preparation; FEH, Designed, performed and analyzed the in vitro electrophysiology experiments, guided the project, Contributed to manuscript preparation; CIDZ, Designed, performed and analyzed the in vivo electrophysiology experiments, Designed, performed and analyzed the two-photon experiments, Guided the project, Wrote the manuscript; MS, Designed, performed and analyzed the in vivo electrophysiology experiments, Designed, performed and analyzed the two-photon experiments, Designed, performed and analyzed the immunohistochemistry, Guided the project, Wrote the manuscript

## Ethics

Animal experimentation: The experiments performed in this study were approved by the local animal ethical committee ('Dier Experimenten Commissie', DEC).

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
