## [Decision Letter]

Thank you for sending your work entitled “Cerebellar modules operate at different frequencies” for consideration at *eLife*. Your article has been favorably evaluated by a Senior editor and 3 reviewers, one of whom is a member of our Board of Reviewing Editors.

The Reviewing editor and the other reviewers discussed their comments before we reached this decision, and the Reviewing editor has assembled the following comments to help you prepare a revised submission.

This is an important study presenting a systematic examination of the physiological properties of Purkinje cells from different cerebellar areas that are referred to as zonal compartments. Cerebellar compartments are fundamental units of circuit organization and they have been studied from many approaches ranging from developmental biology and genetics to imaging and behavior. The powerful in vivo electrophysiology conducted here is superb and this work can be a highly significant contribution to the field.

A wealth of data is presented showing that many properties, including simple spike firing rate, intrinsic excitability, complex spike rate and width, and simple spikes around complex spikes, vary by cerebellar module. One of these properties, simple spike firing rate, can be reduced in some (anterior) modules by blocking TRPC3 channels. The mapping of physiological heterogeneity of the oft-assumed homogenous Purkinje cell population onto different cerebellar regions, and the proposed role for TRPC3 in this difference is potentially quite interesting. However, as written, it could appear that the paper focuses on the stronger conclusion that the differences across regions are determined by zebrin banding. Here the arguments are weaker for two major reasons. 1) The zebrin>mGluR1b>TRPC3 assumption is indirect. 2) The evidence that zebrin identity is the key predictor of Purkinje cell physiology is not convincing and premature. This is especially confusing because the authors eliminate zebrinII itself as the determinant. Please change the presentation to avoid this problem. Below, the reviewers have provided specific suggestions to address these issues.

The reviewers' specific comments follow:

1) The links between zebrin>mGluR1b>TRPC3 are indirect.

The authors acknowledge this, but still use the TRPC3 data as an argument for their zebrin hypothesis. In the Results section the authors discuss the striped expression of several proteins, and included here are references that lead the reader to information supposedly regarding the striped expression of TRPC3. Unless I am missing something, these references do not discuss anything about TRPC3 stripe expression. In fact, throughout the manuscript the authors are a little vague as to what TRPC3 expression looks like. Are there any actually data for this? Please clarify these issues and provide a better set of references, or revise the statement please. As a suggestion, the authors may want to look at the Allen Brain Atlas for the gene expression of TRPC3 in the adult. To me, on the midline sagittal sections it looks like TRPC3 expression is weak in Purkinje cells of lobules VI/VII and X, which are predominately Z+ lobules. This could be helpful to the authors' argument. However, the problem remains that Figure 5 (as all figures except Figure 1) is based on lobule, not actual zebrin identity.

Too much emphasis is placed on mGluR1b given that no experiments address it.

Have the authors looked at what happens to Purkinje cell firing activity when mGluR itself is blocked? I realize that TRPC3 is an attractive target for several different reasons, but much of the argument for looking at this molecule was in fact its link to mGluR1b.

2) The evidence that zebrin identity is the key predictor of Purkinje cell physiology is not convincing as it is currently presented.

- Importantly, everything rests on the analysis in Figure 1–figure supplement 3B, C, E. Without these panels, all of the results in the paper could be due to regional differences that are independent of zebrin. Figures 2, 3, 4 and 5 all use cerebellar lobule as a proxy for zebrin status (based on average zebrin staining).

- Figure 1 only exists to illustrate experiments not presented in a figure. The text description indicates simultaneous recordings from Z+/- pairs, but Z+ n=3 and Z- n=2...?

- The analyses presented in Figure 1–figure supplement 3 are incomplete.

- Figure 1–figure supplement 3B is a non-quantitative “random comparison” that is not useful.

- In general, Figure 1–figure supplement 3 and the accompanying text in the Results section ,the authors focus on demonstrating that Z+/Z- differences exist in all regions, but they do not show us that there are not also differences based on lobule, independent of zebrin identity. In fact, Figure 1-figure supplement 3E appears to show differences in SS rate in Z+ neurons across zones. If there are indeed differences between Purkinje cells in different lobules with the same zebrin identity, how can we conclude that all further observations based solely on lobule (in other words, the entire rest of the paper) depend on zebrin identity, as claimed?

The reviewers' suggestion was as follows:

The simplest way to de-emphasize the (weak according to the data presented) argument that the data have successfully dissociated zebrin identity from physical location would be to re-order the Figures, moving Figure 1 to after Figures 2, 3 and 4. In addition, conclusions on this point should be toned down. Further, for the current Figures 2, 3, 4 and 5, wording like “Z+ or Z- PCs” should be replaced with “PCs in zebrin +/- lobules”. Instead of referring to features that “depend on the zebrin identity”, it would be better to say “correlates with zebrin intensity of the lobule,” or similar. This is necessary if figures are re-ordered, and is a much more accurate interpretation of the data. The TRPC3 data should be the last figure, more to generate a hypothesis of mechanism, with less emphasis on the lobule>zebrin>mGluR1b>TRPC3 arguments that both reviewers found problematic.

It would also be nice to discuss possible differences in Z+ PCs across modules - because they look different in Figure 1–figure supplement 3E, yet there is no direct comparison. What about if something besides zebrin identity were at play here (see comments below)?

Additional comments:

1) Even if zebrin=TRPC3, we would want to know whether blocking TRPC3 affected more than just simple spike rate. Does intrinsic excitability, complex spike rate and width, and simple spikes around complex spikes depend on TRPC3? To what extent are they linked to changes in SS rate? Some of these experiments have already been done and could be analyzed to strengthen the claim that the many physiological features that vary with region share the same cellular basis. And some slice experiments could easily be done in EEAT2-GFP mice (though this too is indirect), to reduce the dependence on lobule as a proxy for zebrin.

2) In the Introduction, the authors refer to the markers as genetic markers. This is a minor point but what the authors actually mean in this context is “molecular markers”. Along the same lines, in the very next sentence it is more appropriate to refer to the zebrins as “the best known of these molecules”. Indeed zebrinII mRNA expression has been studied a while back by Karl Herrup and Richard Hawkes, but this is not what the authors are referring to here.

3) The Results section has some wording issues under the heading “Complex spike characteristics depend on the zebrin identity”. I am not entirely sure what the authors mean here. Please rephrase.

4) Similarly, the first paragraph under the heading “Complex spike characteristics depend on the zebrin identity” is unclear. What are you actually comparing? I think I understand, but this can be reworded for clarity.

5) Figure 1–figure supplement 3, the legend needs some alterations. The authors state that “In addition to classification based on lobules, cerebellar Purkinje cells can be subdivided based on the input they receive into four transverse zones'. They cite Marzban and Hawkes, 2011. A better Hawkes reference for this should be chosen. I suggest the authors cite Ozol et al., 1999 J Comp Neurol. Second, Purkinje cells are subdivided based on 1) their developmental lineage, 2) gene and protein expression patterns, 3) sensitivity to mutations in the anterior-posterior and medial-lateral axis, AND 4) on the inputs they receive. And, in any case what classifies Purkinje cells based on lobules?

---

## [Author Response]

We have now included more experimental data to confirm the link between zebrin-identity and simple spike firing frequency.

*1) The links between zebrin>mGluR1b>TRPC3 are indirect*.

*The authors acknowledge this, but still use the TRPC3 data as an argument for their zebrin hypothesis. In the Results section the authors discuss the striped expression of several proteins, and included here are references that lead the reader to information supposedly regarding the striped expression of TRPC3. Unless I am missing something, these references do not discuss anything about TRPC3 stripe expression. In fact, throughout the manuscript the authors are a little vague as to what TRPC3 expression looks like. Are there any actually data for this? Please clarify these issues and provide a better set of references, or revise the statement please. As a suggestion, the authors may want to look at the Allen Brain Atlas for the gene expression of TRPC3 in the adult. To me, on the midline sagittal sections it looks like TRPC3 expression is weak in Purkinje cells of lobules VI/VII and X, which are predominately Z+ lobules. This could be helpful to the authors' argument*.

This is an interesting, and admittedly, more difficult point. Even though the link between zebrin (as a biomarker) and mGluR1b is strong, in that the distribution of mGluR1b is complementary to that of zebrin (see Mateos et al., 2001), we agree with the reviewers that the connection from mGluR1b to TRPC3 in term of expression is less clear, and other proteins could very well be involved here. In this manuscript we provide evidence supporting a significant contribution of TRPC3 to the difference in simple spike frequency, but there is no evidence, as far as we know, that the expression of TRPC3 is complementary to that of zebrin. We agree that the link between zebrin and TRPC3 is therefore indirect, and want to make clear that we believe that the evidence for the zebrin hypothesis comes from the experimental data, linking PC activity to zebrin identity (including newly added results). We have toned down the text in the Introduction and Discussion sections regarding the link, and clarify that there is no evidence for a pattern in the expression pattern of TRPC3 other than that in the Allen Brain Atlas.

We had indeed noticed the apparent inhomogeneous pattern of TRPC3 expression in the Allen Brain Atlas and we now refer to it in the Discussion. In fact, we tried staining for TRPC3 ourselves, but could not get a conclusive answer to the question whether or not TRPC3 is expressed in bands or not, probably due to poor antibody quality. We have discussed this issue with colleagues from other labs, and they confirmed that the TRPC3 antibody quality is suboptimal. In the end, we were not confident in the results. But, even though we fully agree with the reviewers that it is an interesting question, we do not believe this to be absolutely essential for the current study. If TRPC3 is not expressed in bands, the differential effect of blocking TRPC3 suggests that there are differences in the expression and/or activity of other proteins that are directly linked to TRPC3 or its pathway. This has been shown for several proteins in this cascade, including e.g. IP3R1, PLCbeta3/4, PKCdelta and mGluR1b. This point is now also addressed in the Discussion.

*However, the problem remains that*
Figure 5
*(as all figures except*
Figure 1*) is based on lobule, not actual zebrin identity*.

We now provide additional evidence obtained in vivo *and in vitro* using EAAT4-eGFP mice, that links the difference in firing activity to zebrin rather than lobular location (see answer to point 2). We believe this also strengthens our choice for lobule III as proxy for Z− and lobule X for Z+ Purkinje cells. We would like to point out that not only all data in Figure 1 and the new Figure 2, but also the data regarding complex spikes in the current Figure 4 (former Figure 3) are largely from PCs of which we determined their zebrin identity via staining or two-photon imaging (i.e. new Figure 4 all panels, except C). This is now indicated more clearly in the Results section, in the new Figure 4 and in the legend.

*Too much emphasis is placed on mGluR1b given that no experiments address it*.

*Have the authors looked at what happens to Purkinje cell firing activity when mGluR itself is blocked? I realize that TRPC3 is an attractive target for several different reasons, but much of the argument for looking at this molecule was in fact its link to mGluR1b*.

We agree that, although mGluR1b could be one of the elements, the link between TRPC3 and mGluR1b is indirect. However, we have previously shown that blocking mGluR1 attenuates Purkinje cell simple spike firing frequency in vitro (Coesmans et al., 2003, Figure 1), which fits with the described mechanism.

Even so, we agree with the reviewers that the link between mGluR1b and TRPC3 is indirect, and therefore (and also in response to comments above) we have shifted the focus from mGluR1b to the cascade of proteins, of which mGluR1b is one element, but that also includes IP3R, PKCdelta, etc. he suggestion to block mGluR1 has been considered, but since there are no specific blockers for mGluR1b and because TRPC3 is the channel carrying mGluR1-activation induced current, we focused on TRPC3. To address this comment of the reviewers, we now consistently throughout the text refer to the cascade of proteins related to TRPC3, and restrict the discussion of the possible role of mGluR1b, and the other elements in the cascade, to the discussion.

*2) The evidence that zebrin identity is the key predictor of Purkinje cell physiology is not convincing as it is currently presented*.

*- Importantly, everything rests on the analysis in*
Figure 1*–figure supplement 3B, C, E. Without these panels, all of the results in the paper could be due to regional differences that are independent of zebrin.*
Figures 2, 3, 4 and 5
*all use cerebellar lobule as a proxy for zebrin status (based on average zebrin staining)*.

We think this is an important comment to address and therefore decided to perform additional experiment in vivo and vitro to more thoroughly test our initial claim. We did so using the EAAT4-eGFP mice, making use of the similar expression patterns of EAAT4 and zebrin. We recorded the activity of EAAT4-eGFP positive and negative Purkinje cells in restricted parts of the cerebellum, either in vivo using two-photon imaging or in vitro using a fluorescence microscope. In vivo we recorded 17 cells (Z+: n=8; Z−: n=9) in lobules V-VI and Crus I, and confirmed the difference between Z+ and Z− PCs (Z+, 36.0 ± 15.5; Z−, 75.8 ± 19.5, *t* = 4.618, *p* < 0.001). In addition, we performed recordings in slices of EAAT4-eGFP mice in two areas that we were not able to reach by two-photon imaging. We obtained recordings from 11 cells (7 vs. 4) in lobules II-V and 11 cells (6 vs. 5) in lobules VIII-IX, that again confirm the difference (II-V: Z+, 30.9± 3.4; Z−, 67.5 ± 34.4, *t* = 2.910, *p*= 0.017; VIII-IX: Z+, 35.4 ± 9.2; Z-, 55.1 ± 18.0, *t* = 2.352, *p* = 0.043). These results are now described in a separate Results paragraph and presented in Figure 2 (in vivo), Figure 4 (in vivo) and added to Figure 5, as panel C (in vitro). Together the in vivo and in vitro results cover nearly the entire vermal cortex. The fact that the differences in simple spike frequency between Z+ and Z− PCs persist in each comparison in these new experiments, as well as in the previously presented comparisons per transverse zone (with new two-photon data added, new Figure 2), we believe, argues strongly against their dependence on lobular location, and instead indicates that they correlate with the zebrin identity of the PC.

*-*
Figure 1
*only exists to illustrate experiments not presented in a figure. The text description indicates simultaneous*
*recordings from Z+/- pairs, but Z+ n=3 and Z- n=2...?*

We agree that the original sample was very small. We have performed additional experiments and confirm our earlier findings, now with n=8 for Z+ and n= 9 for Z− in vivo and 2 additional datasets recorded in vitro. To address the question of whether the differences in PC activity are related to lobules or zebrin, we introduce a new Figure 2 with results obtained using the EAAT4-eGFP mice including the example from the former Figure 1, and analyses of Z+ vs. Z− PC activity per transverse zone. Each of these panels indicates that the difference is related to zebrin identity, rather than which lobule the PCs is in.

*- The analyses presented in Figure 1–figure supplement 3 are incomplete*.

*- Figure 1–figure supplement 3B is a non-quantitative “random comparison” that is not useful*.

To give a complete, un-biased overview, we have replaced the random comparisons with the same plot of the unfolded cerebellum, but now with a color coded dot for each recorded and identified PC (new Figure 2–figure supplement 1).

*- In general, Figure 1–figure supplement 3 and the accompanying text in the Results section, the authors focus on demonstrating that Z+/Z- differences exist in all regions, but they do not show us that there are not also differences based on lobule, independent of zebrin identity. In fact, Figure 1-figure supplement 3E appears to show differences in SS rate in Z+ neurons across zones. If there are indeed differences between Purkinje cells in different lobules with the same zebrin identity, how can we conclude that all further observations based solely on lobule (in other words, the entire rest of the paper) depend*
*on zebrin identity, as claimed?*

This is indeed an interesting point. Before the sample sizes were quite low in most cases, but the additional data obtained from EAAT4-eGFP mice allow us to make a better comparison. In the new dataset the simple spike frequency of Z+ PCs in the anterior zone is lower than that of those in the nodular zone. We would like to emphasize that we do not think that, also considering the large variability in the PC activity within Z+ and Z− groups, Z+ or Z− PCs by definition form a homogeneous group. It is, in our view, very well possible there are sub-groups with their own differences, but this is currently outside the scope of this study.

In that respect it is important to note that we only see evidence for within-group differences in the Z+ group, not between subsets of Z− PCs. Yet, it are these putatively more homogenous Z− PCs, of which the activity shifts towards that of the group of Z+ PCs by blocking TRPC3, indicating that the potential differences within the group of Z+ PCs do not affect our conclusions.

We now address this matter in the Results section and refer to it in the Discussion section.

*The reviewers' suggestion*
*was as follows:*

*The simplest way to de-emphasize the (weak according to the data presented) argument that the data have successfully dissociated zebrin identity from physical location would be to re-order the Figures, moving*
Figure 1
*to after*
Figures 2, 3 and 4*. In addition, conclusions on this point should be toned down*.

We believe this is an essential point for our manuscript and have therefore performed additional experiments to obtain more insight in the correlation of PC activity with lobule and/or zebrin identity. These results, described more extensively above, argue against an important contribution of the physical (lobular) location, and strengthen the evidence for the direct relation between zebrin identity and Purkinje cell activity. We understand the reviewer suggestion to change the figure order in order to de-emphasize our argument, but felt that the new datasets we obtained could influence this suggestion. Therefore we consulted the editors via email, inquiring if, based on these new results, we could maintain the original order. We appreciate the opportunity to ask this question directly. Following their advice to proceed as we suggested, we did not change the order of the figures, but instead include new results obtained with EAAT4-eGFP mice in the newly added Figure 2 and new Figure 5. In addition, based on the suggestions of the reviewers, we specifically address this topic in a separate paragraph in the results, and toned down our conclusions in the Discussion.

*Further, for the current*
Figure 2*-5, wording like “Z+ or Z- PCs” should be replaced with “PCs in zebrin +/- lobules”. Instead of referring to features that “depend on the zebrin identity”, it would be better to say “correlates with zebrin intensity of the lobule,” or similar. This is necessary if figures are re-ordered, and is a much more accurate interpretation of the data*.

We agree that the wording should be accurately reflecting the status of the obtained recordings, but we believe there is a misunderstanding. Except for panel 4C, the data in this paragraph describe the characteristics of complex spikes in immunostaining identified Z+ vs. Z− PCs, the same group as described in Figure 1. We now state this more clearly in the text and legend. Panel C is based on random Purkinje cell recordings in different lobules and hemisphere areas, the same as those described in the new Figure 3, and regarding those we do not directly refer to zebrin identity.

We have now added the complex spike activity and regularity in two-photon imaging identified Z+ vs. Z− PCs to this Figure in panel B. Here too, comparable to the results obtained using immunostaining, we found a significantly higher complex spike frequency in Z+ compared to Z− PCs.

*The TRPC3 data should be the last figure, more to generate a hypothesis of mechanism, with less emphasis on the lobule>zebrin>mGluR1b>TRPC3 arguments that both reviewers found problematic*.

The TRPC3 data are only presented in the last figure. We have changed the Introduction, Results and Discussion (as described above) to de-emphasized the link of mGluR1b to TRPC3 and now restrict ourselves where possible to a discussion of the possible link with the cascade that includes several proteins that are expressed in a zebrin-like pattern.

*It would also be nice to discuss possible differences in Z+ PCs across modules - because they look different in Figure 1–figure supplement 3E, yet there is no direct comparison. What about if something besides*
*zebrin identity were at play here (see comments below)?*

We agree that this is a potentially interesting observation. The new data obtained by two-photon imaging identified Z+ and Z− PCs in more anterior parts seem to confirm the impression of the reviewer. Although we demonstrate here that there are relatively large differences in PCs activity between zebrin-positive and -negative modules, we do not want to exclude the possibility that other factors could further subdivide these groups into subgroups (as also suggested above and discussed in relation to new Figure 2),; probably more subtle inter-subgroup differences. These factors could include differences in: 1) the inputs, 2) other proteins or cascades that affect activity or even 3) the relative expression levels of zebrin or proteins in the cascade with similar expression pattern. We now address the possible differences in Z+ PCs in the Results section and include this consideration in the Discussion section.

*Additional*
*comments:*

*1) Even if zebrin=TRPC3, we would want to know whether blocking TRPC3 affected more than just simple spike rate. Does intrinsic excitability, complex spike rate and width, and simple spikes around complex spikes depend on TRPC3? To what extent are they linked to changes in SS rate? Some of these experiments have already been done and could be analyzed to strengthen the claim that the many physiological features that vary with region share the same cellular basis. And some slice experiments could easily be done in EEAT2-GFP mice (though this too is indirect), to reduce the dependence on lobule as a proxy for zebrin*.

In addition to simple spike firing frequency, blocking TRPC3 with selective blocker pyr3 *in vivo* affected several other parameters, including: climbing fiber pause, complex spike frequency and half width. Blocking TRPC3 with pyr3 consistently shifted these parameters in lobule III towards values observed in lobule X without blocker, typically changing significantly compared to lobule III without blocker. In addition, the effect on post-complex spike configuration was interesting, as blocking TRPC3 with pyr3 introduced the configuration types suppression and oscillation in PCs of lobule III, where they do not occur when TRPC3 is not blocked. The effects of genistein were less consistent, presumably due to its less specific nature. These results have now been added as Figure 6—figure supplement 3 and are addressed in the Results section and Discussion section

Since the expression of EAAT4 is practically identical (at least in the vermis) to that of zebrin, we did consider it worthwhile to use the EAAT4-eGFP mice to test whether the differences between Z+ and Z− PCs still exist in Purkinje cells that are virtually side-by-side in the same lobule(s) in coronal cerebellar slices. As described above, we confirmed that the differences pertained both in a set of PCs from lobule II-V and in a separate set from lobule VIII-IX, and included these data (Figure 5).

*2) In the Introduction, the authors refer to the markers as genetic markers. This is a minor point but what the authors actually mean in this context is “molecular markers”. Along the same lines, in the very next sentence it is more appropriate to refer to the zebrins as “the best known of these molecules”. Indeed zebrinII “ expression has been studied a while back by Karl Herrup and Richard Hawkes, but this is not what the authors are referring to here*.

We thank the reviewer for pointing these errors. We have corrected them accordingly.

*3) The Results section has some wording issues under the heading “Complex spike characteristics depend on the zebrin identity”. I am not entirely sure what the authors mean here. Please rephrase*.

We assume the reviewer is referring to the use of the word “coincided” here and have replaced it. The sentence now reads: “This higher firing frequency in lobule III is associated with a higher intrinsic excitability in lobule III compared to lobule X, reinforcing the interpretation that the difference is predominantly intrinsic to Purkinje cells (Figure 4–figure supplement 1).”

*4) Similarly, the first paragraph under the heading “Complex spike characteristics depend on the zebrin identity” is unclear. What are you actually comparing? I think I understand, but this can be reworded for clarity*.

We have attempted to clarify this sentence; it now reads:

“In both *a6-Cacna1a* and *PC-Δγ2* mutant mice the differences in vivo in SS firing frequency between lobules I-III and lobule X were still significant, analogous to normal mice (*a6-Cacna1a*, I-III: 75.1 ± 19.0 Hz, X: 49.1 ± 9.9 Hz, *t* = 3.988, *p* < 0.001; *PC-Δγ2*, I-III: 89.8 ± 14.9 Hz, X: 60.9 ± 15.6 Hz, *t* = 4.876, *p* < 0.001) (Figure 4).”

*5) Figure 1–figure supplement 3, the legend needs some alterations. The authors state that “In addition to classification based on lobules, cerebellar Purkinje cells can be subdivided based on the input they receive into four transverse zones'. They cite Marzban and Hawkes, 2011. A better Hawkes reference for this should be chosen. I suggest the authors cite Ozol et al., 1999 J Comp Neurol. Second, Purkinje cells are subdivided based on 1) their developmental lineage, 2) gene and protein expression patterns, 3) sensitivity to mutations in the anterior-posterior and medial-lateral axis, AND 4) on the inputs they receive. And, in any*
*case what classifies Purkinje cells based on lobules?*

We have replaced the reference according to the reviewer’s suggestion. Regarding the second point, we appreciate the variety of potential ways to subdivide the cerebellar cortex, however, in our current dataset we are restricted by the parameters we determined. We believe there could be a role for diversity in inputs to different parts of the cortex (and we now include this also in the discussion on differences in Z+ PCs), but show that the simple spike frequency differences can be largely explained by intrinsic activity. We appreciate the reviewer’s comment that a subdivision based on lobules has little added value and have removed these panels.